# Impact of COVID-19 Lock-Downs on Nature Connection in Southern and Eastern Africa

Ju-hyoung Lee [1,*], Madalitso Mkandawire [2], Patrick Niyigena [3], Abonisiwe Xotyeni [4], Edwin Itamba [5] and Sylvester Siame [6]

1    Department of Forest Resources, Yeungnam University, 280 Daehak-ro, Gyeongsan 38541, Korea
2    Kasungu District Agriculture Office, Kasungu P.O. Box 17, Malawi; madalitso.mkandawire@agriculture.gov.mw
3    Rwanda Youth in Agribusiness Forum, KG 569 Street, Kigali P.O. Box 7202, Rwanda; pani2101@student.miun.se
4    Park Chung Hee School of Policy and Saemaul (PSPS), Yeungnam University, Gyeongsan 38541, Korea; aboxotyeni@ynu.ac.kr
5    Statistics and Logistic Officer, Sengerema District Council, Mwanza P.O. Box 175, Tanzania; elimu.msingi@sengeremadc.go.tz
6    Forestry Department, Ministry of Green Economy and Environment, Lusaka P.O. Box 50042, Zambia; sylvester.siame@mgee.gov.zm
*    Correspondence: jhlee9@yu.ac.kr; Tel.: +82-53-810-2927

**Abstract:** The response of African countries immediately after the COVID-19 pandemic declaration was rapid and appropriate, with low infections and mortality rates until June 2020. Severe lock-down measures were effective in Africa; however, the reduction in the amount of natural experience influences the quality of life in modern society. This study is conducted as an international comparative study in five African countries on changes in the perception of health recovery and outdoor activities in urban forests during the COVID-19 pandemic. An online survey was conducted with 430 respondents to investigate the relationships between COVID-19 stress, indoor activity, appreciation of greenspaces, perception of health recovery, and use of greenspaces. A structural equation model was used for analysis. The visit frequency and staying time in urban forests after lock-down dramatically decreased, raising concerns about nature-deficit disorder across the target countries after the end of the pandemic. This study confirmed urban dwellers' desire for natural experiences and health recovery during the pandemic and predicts an explosive increase in urban forest utilization after the pandemic has ended.

**Keywords:** COVID-19 in Africa; lock-down; urban forests; health recovery; outdoor recreation

## 1. Introduction

As COVID-19 swept through the world, many observers predicted that African countries would experience a much more deadly epidemic than those in Europe and the United States [1–3]. Because of insufficient hospital beds and fragile health systems, comorbidities such as Ebola, human immunodeficiency virus (HIV), tuberculosis, malnutrition of the young population, overcrowded living environments, etc., were expected to affect infection and mortality negatively [4–6]. However, many African countries have unexpectedly maintained low morbidity and mortality rates through strong lock-down measures in the early stages [1,5,7,8].

There were around 160,000 confirmed COVID-19 cases in Italy and 560,000 cases in the United States in April 2020, compared with approximately 14,000 cases in the African countries. [7].

Domestic and international travel restrictions, curfews, and the closing of offices, schools, and jobs in response to infectious diseases in Africa have been in effect since the days of Lassa Fever and Ebola before COVID-19. In particular, the Ebola virus, which

started in Sierra Leone in March 2014 in Africa and was declared a pandemic in June by the World Health Organization (WHO) is suspected of 28,616 probable and 11,310 deaths in Guinea, Sierra Leone, and Liberia [9]. While an early response to COVID-19 prevented foreign inflow and domestic transmission, the government had enough time to implement strategies such as preparing medical systems for diagnosis, the quarantine of goods, contact tracing, and social distancing [10]. However, a weak health care system and insufficient medical equipment are still threats to combatting diseases in underdeveloped and developing countries in Africa [4,10,11].

The African continent's healthcare and hygiene sectors are weak, including inadequate or outdated healthcare facilities and systems, low public investment in healthcare costs per capita (Europe and the US: over \$2500/capita; most west African countries: below \$50/capita), and a dense population living in poverty and unsanitary living conditions [10]. In the case of the United States, there are 34 intensive care unit beds per 100,000 people compared with only four intensive beds in Kenya [7].

The medical system in Africa has been strengthened through the cooperation and support of international organizations to cope with HIV, malaria, and Ebola treatments, and the African Center for Disease Control was opened and began operating in 2017 [12]. Despite this, most African governments focused on the most effective containment measures in the initial COVID-19 response.

As soon as the COVID-19 outbreak began in Africa, the African Union quickly responded with tiered public health and social measures (PHSM) systems, the most effective tool to combat the fast-spreading epidemic [13]. A clear awareness of the realities of the medical and social health systems enabled governments in Africa to quickly decide on strict responses in the first step of the COVID-19 outbreak [7,13]. At the start of the pandemic, the majority of governments put measures in place such as closing borders, shutting down markets, suspending internal and international flights, and instituting bans on people gatherings [7,11,13].

As a result of the rapid and thorough lock-down in African countries, it was possible to lower the initial infection rate. However, there were side effects such as restrictions on outdoor recreational activities, the interruption of nature experiences, and reductions in the amount of nature experience.

The reward hypothesis is one of the oldest hypotheses in outdoor recreation and leisure research, stating that leisure plays a rewarding role after a hard time [14]. In this study, appreciation of the natural environment and increased outdoor activity in urban forests as a reward for extreme freedom restrictions were established as a research hypothesis.

During the pandemic, a target area survey was conducted on urban populations in the capitals of Malawi, Rwanda, South Africa, Tanzania, and Zambia. This study was conducted to investigate and analyze changes in stress level caused by COVID-19, appreciation of the natural environment, recreational behavior in urban forests, and perceptions of health recovery in a natural setting.

In an online survey on outdoor recreation conducted on American adults in 2020 right after the outbreak of COVID-19, most of the respondents did not want to risk infection due to outdoor recreation during the pandemic, and this perception was based on decreasing overall outdoor recreation [15]. A survey of Canadian adults showed a significant decrease in physical activity after COVID-19 was reported [16], and a study of Vermont residents in the US also found that outdoor recreational activities declined during the pandemic, except hiking and gardening [17]. Urban parks, community centers, recreational centers, and public places of social activity were closed worldwide during COVID-19, and people's work, learning, and leisure activity patterns mainly changed [18].

Recently, research was conducted on overcoming nature-deficit disorder through frequent exposure to the natural environment [19]. The pandemic is decreasing people's nature experience and changing their perceptions of the natural environment due to restrictions on outdoor recreation in the natural environment worldwide.

This study aims to investigate and analyze perceptions of urban forests in limited and disconnected situations and to identify the relationship between the COVID-19-stress experienced by urban residents and their recreational behaviors and perceptions of health recovery in nature. A structural equation model (SEM) research model is used to investigate the perception of nature experience in five African countries during the pandemic era [20]. The research hypotheses are as follows: (i) mental stress caused by restricted face-to-face communication and limited freedom of activity; (ii) increased screen and internet use due to increased indoor living time; (iii) appreciation of urban forest due to (i) and (ii); (iv) actual behavioral changes such as increasing visit frequency; (v) perception of health recovery in the natural environment.

## 2. Materials and Methods

### 2.1. Survey Procedure

The data were collected between April and June 2020, when governments of the five countries began to take countermeasures, including school closures, stay-at-home rules, workplace closures, and restrictions on internal movement, following the WHO pandemic declaration of March 2020 [21]. The target countries were Malawi, Rwanda, South Africa, Tanzania, and Zambia in the southeastern region of Africa, and the people living in the capitals of each country were chosen. Respondents were contacted through email due to restrictions on physical meetings during the survey period [1,5,7,8].

Because of the difficulty of finding respondents in the COVID-19 era, during which social, economic, and human life was abnormal, we secured a list of respondents with cooperative researchers (the co-authors) working in the forestry or environment departments of government agencies in each country. The respondents were selected from groups of individuals who participated in government campaigns or program meetings with the theme of community development, natural landscape, biodiversity conservation, and climate change by the ministry of environment and forestry.

All the participants were informed through consent forms of the research aim, subjects, and ethics at the beginning of the surveys. They knew their right to participate or refuse during the survey questionnaire, such as ending the survey immediately. Excluding errors and insufficient responses, a total of 430 responses were used (Table 1).

**Table 1.** Characteristics of participants and current status in each country.

| Country, Surveyed Area | Forested Area (% of Land Surface) | Population (×1000) | GDP Growth Rates | | | GDP/ People ($) | Survey Respondents | | | | | |
|---|---|---|---|---|---|---|---|---|---|---|---|---|
| | | | | | | | Total (Pers.) | Female (%) | Age (%) | | | |
| | | | '19 | '20 | '21 | | | | ~29 | ~39 | ~49 | 50~ |
| Malawi, Lilongwe | 22,417 sq.km (23.7%) | 19,129 | 5.5 | 0.9 | 2.2 | 636 | 95 | 46.3 | 29.5 | 43.2 | 25.3 | 2.1 |
| Rwanda, Kigali | 2760 sq.km (11.1%) | 12,952 | 10.1 | 3.5 | 6.7 | 797 | 65 | 47.6 | 73.8 | 16.9 | 3.1 | 5.2 |
| South Africa, Pretoria | 170,500 sq.km (14.0%) | 59,308 | 0.2 | −5.8 | 4.0 | 5655 | 94 | 42.5 | 39.4 | 26.6 | 26.6 | 7.4 |
| Tanzania, Dodoma | 457,450 sq.km (51.6%) | 59,734 | 6.3 | 2.0 | 4.6 | 1076 | 74 | 37.8 | 12.2 | 48.6 | 39.2 | 0.0 |
| Zambia, Lusaka | 448,140 sq.km (60.2%) | 18,383 | 1.5 | −3.5 | 2.3 | 985 | 102 | 51.9 | 8.8 | 39.2 | 48.0 | 3.9 |

(Source: Worldbank 2020, IMF 2020, and survey results in target areas).

### 2.2. Survey Method and Statistical Analysis

To explore the research hypotheses, we applied developed survey questions on the perception of nature experience in five African countries during the pandemic [20]. The survey questionnaires were revised into 12 questions in 4 categories: COVID-19 Stress level, indoor activity increase, enhanced appreciation of urban forests and natural environment, and agreement with health recovery in nature (Table 2) [20]. The participants responded using a 5-point scale.

**Table 2.** Interview questionnaire's reliability by categories.

| Category | Items (Since COVID-19, … ) | Var. | Cronbach's $\alpha$ | | | | |
|---|---|---|---|---|---|---|---|
| | | | **MA** | **RW** | **SA** | **TA** | **ZA** |
| Stress level | I am unsatisfied with the restricted daily life (activity, visitation, work, school). | ST1 | 0.826 | 0.860 | 0.831 | 0.801 | 0.878 |
| | I am unsatisfied with limited communication opportunities with other people. | ST2 | | | | | |
| | It has decreased communication with other people (except family). | ST3 | | | | | |
| | It has increased the communication with my family. * | ST4 | | | | | |
| Indoor activity | I don't have enough leisure activities. | ID1 | 0.821 | 0.792 | 0.803 | 0.841 | 0.862 |
| | The screen time for visiting websites and watching TV has increased. | ID2 | | | | | |
| | I prefer online activities to be offline (shopping, learning, communicating, etc.). | ID3 | | | | | |
| Appreciation of urban forests | I get a positive feeling when I visit urban forests. * | PR1 | 0.793 | 0.774 | 0.836 | 0.810 | 0.775 |
| | It became more challenging to go to the outdoor natural environment in urban forests. | PR2 | | | | | |
| | I love to experience the outdoor nature around me, particularly during the pandemic. * | PR3 | | | | | |
| Perception of health recovery in nature | Through nature experience in the urban forest, I felt my mental recovery. * | HE1 | 0.786 | 0.797 | 0.775 | 0.852 | 0.784 |
| | Through nature experience in the urban forest, I felt my body recover. * | HE2 | | | | | |

Note. * reversed items (Questions were adapted from [20]).

The questionnaires were translated and used for field surveys in each country, and the 12 questions suitable for statistical significance were finally selected. The internal consistency reliability of each country was analyzed reliably with Cronbach's alpha scores greater than 0.65 [22].

Behavior change in urban forest experience was analyzed with the public health and social measures (PHSM) Index provided by the Africa Center for Disease Control and Prevention (CDC) by comparing outdoor recreation behaviors before and after COVID-19. Tiered PHSM systems are a core component of effective COVID-19 preparedness, response, and risk communication [13]. The WHO classifies the PHSM Severity Index into six indicators: (1) the earing of face masks, (2) school closures, (3) office, business, and institution closures, (4) bans on people gathering, and (5) bans on domestic and (6) international travel and movement [23].

Before and after the COVID-19 lock-downs, outdoor recreation was quantified on two five-point scales for frequency, duration of greenspaces visits, and transportation time. Visit frequency measured the average frequency of visits to urban forests and urban green areas: (5) every day; (4) 2–3 times/week; (3) 1 time/week; (2) 1 time/month; and (1) rarely. Staying time measured the average length of staying time: (5) a day; (4) 4 h; (3) 2 h; (2) 1 h; and (1) 30 min. Transport time was the average travel time to the destination: (5) less than 15 min.; (4) less than 30 min.; (3) less than 1 h; (2) 1~2 h; and (1) more than 2 h.

Appreciation of urban forest, or appreciation of a natural environment in an urban setting, was affected by exogenous variables during the pandemic. Stress level was an exogenous variable related to observed interpersonal communications and relationship variables. Indoor activity was an exogenous variable affected by observational variables, including (on- or offline) screen time and visiting websites due to increase staying time at home. Outdoor recreation included the behavior of the outdoor activity, while health recovery (in nature) was related to the attention to nature-based human health during the pandemic era. (Figure 1) [20].

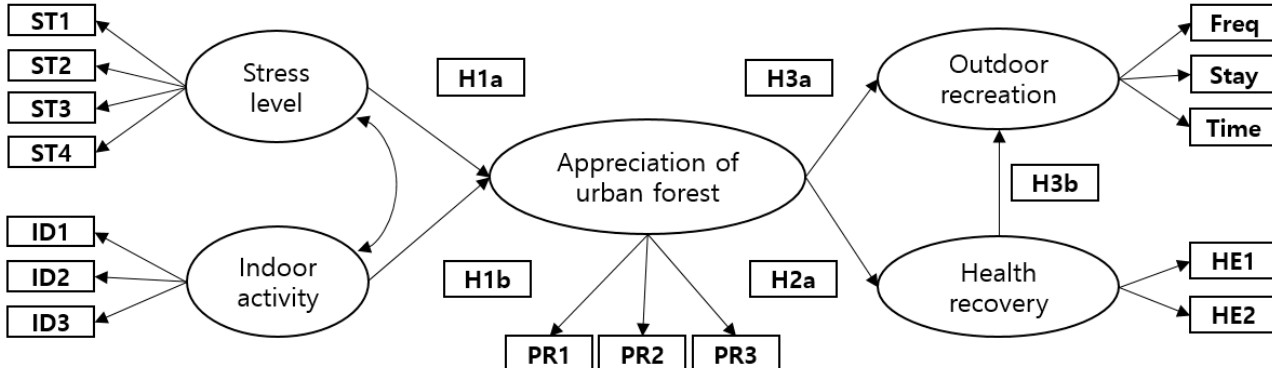

**Figure 1.** Structural equation modeling diagram (Adapted from [20]). Freq: visit frequency to urban forests; stay: staying time in urban forests; time: transport time to natural areas in urban settings. H1: respondent's appreciation of urban forests comes from stress level and indoor activity; H2: people's perception of health recovery during the COVID-19 pandemic comes from appreciation of urban forests; H3: behavior changes in outdoor activity come from both appreciation of urban forests and expectations for health recovery.

The collected data were analyzed using an analysis of variance (ANOVA) approach to examine the differences in each country and with structural equation modeling (SEM) to explore the relationships between variables. SPSS 25 and AMOS 25 were used for statistical analysis, and confirmatory factor analysis (CFA) was conducted to estimate the measured variables' fitness and each type's characteristics [24]. Path analysis was performed to test the influence relationships between the measured variables. Construct reliability, average variance extracted, and covariance values for the measured variables were calculated to examine the reliability and validity of the research model. To determine the intervening effect's statistical significance, we used a Sobel test.

## 3. Results

### 3.1. Comparative Changes in Perception and Behavior

The question of this study began at these points: Does increasing COVID-19 stress among urban dwellers with limited contact change their perceptions of outdoor activities and nature experiences? Do the country-specific PHSM measures identified by the Africa CDC make changes that increase or decrease the amount of natural experience of urban dwellers? This study was conducted to answer these questions.

Changes in outdoor recreation behavior before and after COVID-19 in each country were collected through the PHSM index (Table 3) [13]. The changed outdoor recreation behaviors pre- and post-COVID-19 results are presented in Table 4.

In the research SEM model, the increases in stress and indoor activities led to a rise in the appreciation of urban forests. The increased interest in health under a pandemic led to a perception of health recovery in the natural environment and outdoor activity (Figure 1). However, due to the governments' lock-down measures (Table 3), the total amount of natural experiences, such as frequency of nature experience and length of stay, of respondents in all the countries decreased significantly during the COVID-19 pandemic (Table 4). The difference in the frequency of visits before and after was verified at a high significance level.

These results were consistent with the indicators of schools closing, canceled public events, restrictions on gatherings, international travel controls, and facial coverings and mask-wearing that were common in PHSM systems during the period (Table 3). The staying time in the visited urban forests decreased in all except Malawi and was statistically significant by t-test. In other words, the visit frequency to urban forests and the time spent in urban forests both decreased during COVID-19, indicating that the total amount of city residents' natural experiences reduced significantly (Table 4). The transport time for urban

forest visits increased in Malawi and Tanzania and decreased in Rwanda, South Africa, and Zambia. However, since statistical significance was not secured (South Africa and Zambia) or there were mixed cases of increase (Malawi) or decrease (Tanzania), the circumstances of each country were different, so an integrated interpretation was difficult.

**Table 3.** Comparison PHSM index between pre-and post-COVID-19 outbreak (Resource: Africa CDC).

| Country | Malawi | Rwanda | South Africa | Tanzania | Zambia |
|---|---|---|---|---|---|
| Month | January–June | January–June | January–June | January–June | January–June |

Notes. School: school closings; Workplace: workplace closings; Public events: canceled public events; Gathering: restrictions on gatherings; Transport: closed public transport; Stay at home: stay at home requirements; Movement: restrictions on internal movement; Travel abroad: international travel controls; Mask: facial coverings or mask-wearing requirements (Resource: Africa CDC). PHSM index: no data; recommended; subnational only; national.

**Table 4.** Changed outdoor recreation behavior during COVID-19 pandemic.

| | Visit Frequency | | | Staying Time | | | Transport Time | | |
|---|---|---|---|---|---|---|---|---|---|
| Month | January | June | t-value | January | June | t-value | January | June | t-value |
| Malawi | 2.61 | 2.15 | 3.009 ** | 3.07 | 3.51 | −2.993 ** | 3.66 | 3.96 | −3.243 ** |
| Rwanda | 3.66 | 2.38 | 6.022 *** | 3.48 | 2.48 | 4.309 *** | 3.49 | 3.00 | 2.075 * |
| South Africa | 2.77 | 2.34 | 4.535 *** | 2.71 | 1.96 | 5.647 *** | 3.06 | 2.98 | 1.051 |
| Tanzania | 3.44 | 2.00 | 8.021 *** | 2.97 | 2.00 | 7.083 *** | 3.16 | 3.49 | −2.042 * |
| Zambia | 3.32 | 2.59 | 4.630 *** | 3.25 | 2.67 | 4.338 *** | 3.48 | 3.27 | 1.902 |
| Average | 3.11 | 2.30 | 11.110 *** | 3.08 | 2.55 | 7.214 *** | 3.37 | 3.35 | 0.350 |

Notes. *p*-value: * $p < 0.05$, ** $p < 0.01$, *** $p < 0.001$. All questionnaires were measured on a 5-point scale from 5 (every day) to 1 (rarely) for visit frequency, from 5 (a day) to 1 (less than 30 min) for staying time, and from 5 (less than 15 min) to 1 (more than 1 h) for transport time.

The COVID-19 stress was highest in Tanzania, and for the indoor activity category, Zambia and Malawi responded highly, reflecting an increase in indoor staying time due to the earliest restrictions implemented in Malawi (Table 5). South Africa showed the highest appreciation of urban forests during the pandemic.

Most of the respondents answered that it was newly recognized that the natural environment helped people recover their psychological and physical health during the pandemic, and Tanzania showed the highest number. Respondents from all the countries answered that the effect of physical health recovery was greater than that of psychological health recovery, in contrast to an Asian study that showed high responses to psychological health to the same question [20]. The visit frequency (T = 11.110; $p = 0.000$) and duration of visits (T = 7.214; $p = 0.000$) to urban forests significantly decreased due to lock-down measures; these data did not match the research hypothesis. Hypothesis H3 was rejected and dropped from SEM diagram.

**Table 5.** Survey results overview: appreciation and perception of nature in five countries.

| | Stress Level | Indoor Activity | Appreciation of Urban Forests | Perception of Health Recovery |
|---|---|---|---|---|
| Malawi | 3.74 ± 0.39 | 4.10 ± 0.28 [a,b] | 3.51 ± 0.36 | 3.81 ± 0.40 [b] |
| Rwanda | 3.70 ± 0.42 | 3.89 ± 0.46 [b] | 3.62 ± 0.45 | 3.80 ± 0.47 [b] |
| South Africa | 3.82 ± 0.48 | 3.80 ± 0.46 [b] | 3.74 ± 0.43 | 3.93 ± 0.42 [a,b] |
| Tanzania | 3.93 ± 0.38 | 3.99 ± 0.40 [a,b] | 3.61 ± 0.48 | 4.23 ± 0.40 [a] |
| Zambia | 3.57 ± 0.44 | 4.29 ± 0.39 [a] | 3.54 ± 0.48 | 4.05 ± 0.48 [a,b] |
| F-value | 2.042 | 4.895 | 0.968 | 3.450 |
| *p*-value | 0.088 | 0.001 | 0.425 | 0.009 |

Notes. [a,b] same letter not significant ($\alpha = 0.05$) Scheffe's post-hoc test; MANOVA Wilks' lambda = 4.033; $p < 0.000$; partial eta squared = 0.04.

### 3.2. Confirmatory Factor Analysis

A CFA was performed to test the validities of the variables (Table 6). Before confirming the CFA model, the variables' redundancies were tested. All the items were calculated with high $R^2$ values significantly by good concentration validity.

**Table 6.** Confirmatory factor analysis results.

| Category | | Variables | Malawi | Rwanda | South Africa | Tanzania | Zambia |
|---|---|---|---|---|---|---|---|
| Stress level | ß-coeff. | ST1 | 0.708 | 0.912 | 0.675 | 0.875 | 0.852 |
| | | ST2 | 0.695 | 0.704 | 0.762 | 0.958 | 0.967 |
| | | ST3 | 0.683 | 0.834 | 0.828 | 0.509 | 0.788 |
| | | ST4 | 0.890 | 0.716 | 0.718 | 0.486 | 0.616 |
| | | CR | 0.840 | 0.855 | 0.813 | 0.829 | 0.879 |
| | | AVE | 0.571 | 0.599 | 0.522 | 0.569 | 0.651 |
| Indoor activity | ß-coeff. | ID1 | 0.753 | 0.573 | 0.554 | 0.720 | 0.762 |
| | | ID2 | 0.787 | 0.716 | 0.913 | 0.993 | 0.884 |
| | | ID3 | 0.792 | 0.910 | 0.831 | 0.753 | 0.827 |
| | | CR | 0.890 | 0.735 | 0.797 | 0.879 | 0.879 |
| | | AVE | 0.729 | 0.509 | 0.577 | 0.713 | 0.709 |
| Appreciation of urban forests | ß-coeff. | PR1 | 0.631 | 0.628 | 0.864 | 0.674 | 0.756 |
| | | PR2 | 0.611 | 0.849 | 0.779 | 0.678 | 0.770 |
| | | PR3 | 0.998 | 0.735 | 0.757 | 0.976 | 0.673 |
| | | CR | 0.833 | 0.760 | 0.839 | 0.800 | 0.731 |
| | | AVE | 0.635 | 0.519 | 0.635 | 0.580 | 0.506 |
| Health recovery perception | ß-coeff. | HE1 | 0.729 | 0.894 | 0.772 | 0.942 | 0.742 |
| | | HE2 | 0.937 | 0.743 | 0.830 | 0.794 | 0.869 |
| | | CR | 0.843 | 0.787 | 0.732 | 0.890 | 0.851 |
| | | AVE | 0.732 | 0.651 | 0.578 | 0.803 | 0.742 |
| Model fit summary | | Chi2 | 103.768 | 60.609 | 92.394 | 137.957 | 136.846 |
| | | P-level | 0.000 | 0.105 | 0.000 | 0.000 | 0.000 |
| | | GFI | 0.860 | 0.865 | 0.877 | 0.791 | 0.822 |
| | | AGFI | 0.773 | 0.781 | 0.801 | 0.740 | 0.790 |
| | | NFI | 0.810 | 0.858 | 0.840 | 0.846 | 0.815 |
| | | IFI | 0.888 | 0.967 | 0.916 | 0.918 | 0.872 |
| | | TLI | 0.840 | 0.952 | 0.910 | 0.894 | 0.881 |
| | | CFI | 0.884 | 0.965 | 0.913 | 0.912 | 0.916 |
| | | RMSEA | 0.071 | 0.054 | 0.056 | 0.051 | 0.063 |

Notes. ß-coeff.: standardized coefficient, CR: construct reliability, AVE: average variance extracted, P-level: probability level.

The exogenous variables were composed of four variables of stress level and three of indoor activity. The endogenous variables consisted of two variables of health recovery and three of appreciation of urban forests.

The construct reliability (CR) and average variance extracted (AVE) were calculated to test discriminant validity. The validity of the AVE range value was: very good (>0.7), acceptable (0.7~0.5), and not acceptable (<0.5). If the CR value was above 0.7 and the AVE was above 0.5, the data could be determined to have convergent validity [25,26]. The research CFA model was confirmed with high CR and AVE values for internal consistency and convergence validity, which indicated a good data fit. The model fit test by each country is shown in Table 6. The model was suitable when the chi-square value was small and the probability value was considerable ($p > 0.10$). The indices of model fit by chi square were not acceptable for Malawi (103.768, $p = 0.000$), South Africa (92.394, $p = 0.000$), Tanzania (137.957, $p = 0.000$), and Zambia (136.846, $p = 0.0000$), but they were acceptable for Rwanda at 60.609 ($p = 0.105$).

The minimum and maximum indices of the five countries were goodness-of-fit index (GFI) = 0.791~0.877 and adjusted goodness-of-fit (AGFI) = 0.773~0.801. The GFI and AGFI statistics ranged between 0 and 1, and the recommended values, a good fit, were more than 0.90. The GFI value of 0.791~0.877 was less than 0.9 due to the relatively small sample size [27,28].

The normed fit index (NFI) was 0.810~0.858. The normed fit index (NFI) was used as an alternative to CFI, but one did not require the chi-square test, and the range was from 0 to 1, with 1 = perfect fit [29]. The incremental fit index (IFI) = 0.872~0.967, the Tucker–Lewis index (TLI) = 0.840~0.952, and the comparative fit index (CFI) = 0.884~0.965. The CFI value was close to 0.9, which showed a relatively good fit [30]. The TLI is relatively independent of sample size, and a TLI value approaching 1 indicates a good fit [28,29,31,32].

The root mean square error of approximation (RMSEA) = 0.051~0.071. RMSEA is an index of the difference between the observed covariance matrix per degree of freedom and the hypothesized covariance matrix, which denotes the model [25,33]. A good model fit by RMSEA was recommended as smaller than 0.06, in general [34], from 0.05 to 0.07 are acceptable [35], 0.08 to 0.1 are marginal, and values above 0.1 are poor [36]. The CFA model fit was evaluated to be acceptable (Table 6).

### 3.3. Structural Equation Modeling for the Research Hypotheses

SEM was conducted to examine the hypothesis of the relationship between COVID-19 stress and changed perceptions of health in the urban forests of five African countries' people (Table 7, Appendix A (Table A1)). The model fit indices by chi square were not acceptable for Malawi ($\chi^2 = 119.963$, $p = 0.000$), South Africa ($\chi^2 = 130.390$, $p = 0.000$), Tanzania ($\chi^2 = 151.270$, $p = 0.000$), and Zambia ($\chi^2 = 140.689$, $p = 0.000$), but they were acceptable for Rwanda at $\chi^2 = 64.672$ ($p = 0.079$). The different model fit test results are given in Table 7 and Appendix A (Table A1). The first hypothesis of the relationship of stress level to appreciation of urban forests as deterministic variables was examined. The hypothesis of a positive effect of stress level on urban forest appreciation was accepted in Rwanda, South Africa, and Zambia with standardized coefficients of 0.419, 0.363, and 0.375, respectively (t = 3.081, 3.175, and 3.281, respectively; $p < 0.01$). The level of the symbolic meaning of increasing stress in the COVID-19 pandemic was significantly related to the appreciation of urban forests.

However, the hypothesis of the significant effect of indoor activity, which represented the changed indoor-oriented lifestyle of increased screen-watching time (internet-accessed digital device tablet-PC, smartphone, etc.) to appreciation of urban forests, was supported in two countries of Rwanda and Zambia, with standardized coefficients of 0.447 (t = 2.787, $p < 0.01$) and 0.285 (t = 2.006, $p < 0.05$), respectively.

The second hypothesis of the relationship of health recovery perception to appreciation of urban forests was examined. Except in Tanzania, the appreciation of urban forests and nature experiences was significantly related to recovering health in nature ($p < 0.01$~$0.001$). The results supported the hypothesis that the appreciation of urban forests and nature experiences caused nature-related health recovery perceptions in Rwanda, South Africa, and Zambia.

**Table 7.** Research hypotheses tested by SEM model fit.

| Hypothesis: Direction | | Malawi | Rwanda | South Africa | Tanzania | Zambia |
|---|---|---|---|---|---|---|
| H1a: Stress level → | Appreciation of urban forests | Reject | **Accept** | **Accept** | Reject | **Accept** |
| H1b: Indoor activity → | Appreciation of urban forests | Reject | **Accept** | Reject | Reject | **Accept** |
| H2: Appreciation of urban forests → | Perception of health recovery | **Accept** | **Accept** | **Accept** | Reject | **Accept** |

Because of bans on direct interpersonal contact and increased indoor time, in relation to COVID-19 stress, it was observed that the appreciation of urban forests was enhanced in Rwanda and Zambia. The increased appreciation of nature experiences in urban forests during the COVID-19 pandemic was related to the perception of nature-based health recovery in Malawi, Rwanda, South Africa, and Zambia.

The appreciation of urban forests intervened between stress level, indoor activity, and health recovery. A Sobel test was conducted on the research model, assuming that either stress level or indoor activity affected health recovery perception (Table 8) [37,38].

**Table 8.** The indirect effect of explanatory variables on health outcomes evaluated with the Sobel test.

| Indirect Effect | | Malawi | Rwanda | South Africa | Tanzania | Zambia |
|---|---|---|---|---|---|---|
| Stress level → Urban forest → Health | Z-value | 0.850 | **2.363** | 1.909 | 0.152 | **2.576** |
| | *p* | 0.197 | **0.009** | **0.028** | 0.439 | **0.004** |
| Indoor activity → Urban forest → Health | Z-value | 0.657 | **2.226** | 0.667 | 0.713 | **1.806** |
| | *p* | 0.255 | **0.013** | 0.252 | 0.237 | **0.035** |

Notes. Urban forest: appreciation of urban forests, Health: perception of health recovery.

Appreciation of urban forests acted as an intervening variable between stress level and health recovery perception in Rwanda ($p = 0.009$), South Africa ($p = 0.028$), and Zambia ($p = 0.004$). The second test showed that appreciation of urban forests intervened between indoor activity and health recovery perception in Rwanda ($p = 0.013$) and Zambia ($p = 0.035$). The Sobel test results showed that the appreciation of urban forests in Rwanda and Zambia played a mediating role between stress level during the COVID-19 pandemic and the perception of health recovery.

We found that the stress level and indoor activities affected the appreciation of urban forests associated with the perception of health recovery through research model examination. Thus, the research hypotheses of H1a, H1b, and H2a were fully supported in Rwanda and Zambia.

## 4. Discussion

### 4.1. Distancing from Nature during Lock-Down Measures

The tendency of a decreasing amount of nature experience during the COVID-19 pandemic and increasing stress levels was observed in this study. We studied the bases of these phenomena from limited interpersonal communication during lock-down. In the world after the pandemic declaration, people's stress increased due to a series of government regulations on face-to-face contact, and residents' daily lives in densely populated cities meant disconnection and isolation. In particular, underdeveloped countries imposed strong lock-down measures to prevent the collapse of the health system due to large-scale infection, and the majority of people with weak financial status faced difficulties due to income reduction, food insecurity, and threats to their livelihoods [5,9,10,20,39].

In Africa, where experience in coping with diseases such as tuberculosis, HIV, Ebola, and H1N1 influenza has been accumulated, immediate and strong lock-down policies were implemented after the pandemic. In the five countries of this study (Malawi, Rwanda, South Africa, Tanzania, and Zambia), after the pandemic declaration, strong containment policies were implemented against schools, public gatherings, group contact, travel, and the internal movement of people [7,8,10,11,13].

We found increased stress levels due to limited interpersonal communication during lock-down. In the United States, Poland, Canada, and Australia, compared to the pre-COVID-19 period people complained of psychological disorders such as isolation, depression, helplessness, and stress due to significantly increased indoor dwelling time, limited communication, and insufficient outdoor activities [40–43]. The increase in indoor life has been indicated by passive and low-movement habits such as reading, pc games, tablets, and smartphones [44–49]. Despite interest in various indoor exercises, including yoga, the total amount of activity tended to decrease [50].

Quantitative changes in outdoor recreation activities during the COVID-19 pandemic have been reported worldwide, including in the US [15–17,51]. In this study, a decrease in the frequency of visits to urban forests was observed in five countries.

In a long-term survey of 64,000 people in the United States, despite national and state parks closing and recreation program stoppage due to the federal government's COVID-19 prevention policy, visits to the forest increased [51]. It was reported that the frequency of outdoor recreation among American citizens increased by 43% during the COVID-19 pandemic [52]. In Oslo's urban green areas, pedestrian activity increased during the pandemic, and it was identified that urban nature acted as an escape from various aspects of restriction stress [53].

The frequency of visits to the natural environment has been identified as an essential factor influencing nature experience and outdoor recreation [54–56]. In a comparative study of urban residents in three Asian countries with pro-environmental attitudes [57], the perception and behavior of urban forest visitors in three western European countries for health recovery [58], and a comparative analysis of visitors to urban forests in Korea and Germany with the theme of nature experience aversion, visit frequency was always determined as an important influencing factor [19]. The decrease in the frequency of visits to urban forests after lock-down observed in this study was predicted as a factor that could increase COVID-19 stress and lower the quality of life under the pandemic.

### 4.2. Perception of Recovery in Human Health in Nature during the COVID-19 Pandemic

We predicted that the stress caused by COVID-19 was predicted to express an increase in appreciation of urban forest experiences and affect behavioral changes, such as increased outdoor activities and the perception of health recovery in urban forests. The research hypothesis assumed that the flow of increased interest in health after the pandemic would lead to health recovery perception through natural experiences. All five countries showed very high responses to the idea of the recovery of human health in nature.

The H2 hypothesis of the SEM model assumed a relationship between the appreciation strengthened during the pandemic and the perception of health recovery in nature, and this was supported in four countries, all except Tanzania. As a result, factors such as indoor lifestyle, communication disconnection, and stress in Rwanda and Zambia influenced the perception of health recovery in nature with appreciation of urban forest experiences as a parameter.

Many previous studies have examined health promotion in forests and nature [40–43,59–68]. In a study on the perception of psychological and physical health promotion effects in urban forests conducted in Berlin (Germany), Vienna (Austria), and Zurich (Switzerland), the health recovery function in urban forests was affirmed [58]. Electoencephalography (EEG) analyses have suggested that the alpha wave (the conscious and relaxed brain status with a frequency of 8 to 13 hertz), which is increased during relaxation, has also been observed to increase when the people are exposed to a forest environment, thereby af-

firming that forest experiences have a therapeutic effect and a significant impact on stress reduction [41,42,69,70]. In studies on human physiological substances, the cortisol hormone, detected at a high concentration when people are in a stressed state, has shown a statistically significant decrease when subjects are exposed to a forest environment, thus supporting forest experience programs as a physiological treatment therapy [71–73].

The effectiveness of trekking in forests to promote people's physical health, such as muscular endurance, bone density, and cardiorespiratory function, was studied by planning an appropriate exercise load considering the slope and distance of forest trails in mountainous areas. This has been named "terrain therapy" as a treatment therapy [71,74–77]. The recovery of human health in a forest environment has been shown through research on its effects on mental health treatment, such as alcoholism, gambling addiction, and gaming addiction, and on the improvement of various psychological disorders, such as interpersonal stress, depression, and obsession disorder [41,61,78–82]. In the pandemic, urban green spaces were also newly defined as areas important for life that maximize natural human resilience [83–85].

The results of this study, which demonstrated the relationship between the appreciation of urban forests during the pandemic and the perception of health recovery, are in line with the results of studies in the United States and Europe that reported an increase in the amount of outdoor activity to maintain health [51,86].

### 4.3. Study Limitations

This study had several research limitations that need to be improved in further studies. First, it is necessary to verify whether the sample is representative demographically, geographically, historically, and culturally because the sample collected during the international comparative study was assumed to represent the research target country and region [87–89]. Of course, this study surveyed the citizens of the capital of each country.

Still, in a country where the capital and the provinces have economic, social, and cultural differences, residents of the capital may not represent residents of the entire country. This is another typical problem in a cross-national study of the sample homogeneity issue at the same time. This includes whether the sample group of this study, which was smaller under the pandemic situation, could represent the population. To compensate for these weaknesses, it is necessary to deepen the group composition and interpretation of research results by including sociological or anthropological factors in future studies.

Despite briefly mentioning a decrease in GDP growth rates in five countries before and after the COVID-19, the interpretation of the post-COVID-19 socioeconomic and industrial impact was insufficient. COVID-19 caused severe food insecurity, as well as the extreme decline and partial collapse of tourism, the financial sector, and education systems [90–94].

Considering that macro-and micro-socioeconomic impacts affect even the political arena [9,95], an integrated interpretation of the economic situation and the pandemic is necessary for future research.

After the lock-down, there was an apparent decrease in citizens' urban forest use rates, so we had to revise the SEM model. Although the high appreciation of urban forests and the will to recover health in nature were analyzed, it was impossible to integrate and interpret observed behavioral changes, such as visit frequency and stay time. An analysis of the trend of behavioral change after the pandemic is lifted is necessary as a pre-post-follow-up survey.

### 5. Conclusions

This study aimed to investigate the relationship between urban forest appreciation and health recovery perception with stress, outdoor recreation, and lock-down measures in most but not all of the investigated countries during the COVID-19 pandemic through an SEM model. The study results showed that, in Rwanda, South Africa, and Zambia, COVID-19 stress led to an appreciation of urban forests and was related to health recovery perception. However, at the same time, social distancing from nature due to COVID-19

lock-down rules appeared to decrease the amount of natural experience, such as the visit frequency and stay time.

As the research results showed, appreciation of the natural environment in urban forests developed into high expectations for health recovery in nature. The role and function of the urban forest that has worked so far should be strengthened as a health recovery place that can overcome the stress of COVID-19.

The COVID-19 lock-down measures were a response to a crisis of humanity, and they reinforced the desire to experience nature and restore nature-based health. The results of this study can serve as a basis for predicting the expectation for health recovery and healing activities in urban forests after the end of the pandemic.

**Author Contributions:** Conceptualization, J.-h.L.; methodology, J.-h.L.; software, J.-h.L.; validation, J.-h.L.; formal analysis, J.-h.L.; investigation, M.M., P.N., A.X., E.I., and S.S.; resources, M.M., P.N., A.X., E.I., and S.S.; data curation, M.M., P.N., A.X., E.I., and S.S.; writing—original draft preparation, J.-h.L.; writing—review and editing, J.-h.L.; visualization, J.-h.L.; project administration, J.-h.L. All authors have read and agreed to the published version of the manuscript.

**Funding:** This research received no external funding.

**Institutional Review Board Statement:** Not applicable.

**Informed Consent Statement:** Not applicable.

**Data Availability Statement:** Not applicable.

**Acknowledgments:** The authors would like to thank the Park Chung Hee School of Policy and Saemaul (PSPS), Yeungnam University, Korea, for building the research cooperation relationship with five governments of African countries.

**Conflicts of Interest:** The authors declare no conflict of interest.

## Appendix A

**Table A1.** Research hypotheses tested by SEM model fit.

| Hypothesis: Direction | | | Malawi | Rwanda | South Africa | Tanzania | Zambia |
|---|---|---|---|---|---|---|---|
| H1a: Stress level → | Appreciation of urban forest | Estimate | 0.106 | 0.419 | 0.363 | 0.027 | **0.375** |
| | | S.E. | 0.118 | 0.136 | 0.152 | 0.174 | 0.114 |
| | | CR | 0.898 | 3.081 | 3.175 | 0.002 | 3.281 |
| | | *p* | 0.497 | 0.002 | 0.001 | 0.998 | 0.001 |
| | | Result | Reject | Accept | Accept | Reject | Accept |
| H1b: Indoor activity → | Appreciation of urban forest | Estimate | 0.091 | **0.447** | 0.114 | 0.314 | **0.285** |
| | | S.E. | 0.134 | 0.160 | 0.167 | 0.215 | 0.142 |
| | | CR | 0.679 | 2.787 | 0.680 | 1.463 | 2.006 |
| | | *p* | 0.497 | 0.005 | 0.496 | 0.143 | 0.045 |
| | | Result | Reject | Accept | Reject | Reject | Accept |
| H2: Appreciation of urban forest → | Perception of health recovery | Estimate | **0.454** | **0.571** | 0.563 | 0.067 | **0.402** |
| | | S.E. | 0.172 | 0.155 | 0.177 | 0.082 | 0.097 |
| | | CR | 2.647 | 3.695 | 3.175 | 0.812 | 4.123 |
| | | *p* | 0.008 | 0.000 | 0.001 | 0.417 | 0.000 |
| | | Result | Accept | Accept | Accept | Reject | Accept |
| Model fit test | | Chi2 | 119.963 | 64.672 | 130.390 | 151.270 | 140.689 |
| | | *p*-value | 0.000 | 0.079 | 0.000 | 0.000 | 0.000 |
| | | GFI | 0.838 | 0.859 | 0.838 | 0.769 | 0.812 |
| | | AGFI | 0.747 | 0.781 | 0.747 | 0.639 | 0.707 |
| | | NFI | 0.780 | 0.848 | 0.774 | 0.722 | 0.810 |
| | | IFI | 0.859 | 0.961 | 0.847 | 0.795 | 0.869 |
| | | TLI | 0.807 | 0.946 | 0.792 | 0.720 | 0.822 |
| | | CFI | 0.854 | 0.959 | 0.843 | 0.788 | 0.865 |
| | | RMSEA | 0.122 | 0.048 | 0.052 | 0.167 | 0.057 |

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
