# Peer review of "Impact of COVID-19 Lock-Downs on Nature Connection in Southern and Eastern Africa"

_land, doi:10.3390/land11060872_

Round 1

Reviewer 1 Report

This is an interesting study that examines residents’ preferences for urban forests across five African countries. The reviewer lauded the efforts made by authors to conduct such a study that compares the role of urban forests in people’s daily lives during the harsh times of COVID-19 for the five African countries. The manuscript has merits for publication in Land. That said, the following concerns need to be addressed.

1.  Introduction

While authors provided a good introduction of COVID-19 outbreak in Africa, it lacks a good transition from that introduction to the use of urban forests and associated benefits. It would read better if the role that urban forests played in providing health benefits, especially during the COVID-19 pandemic, can be described. (around p. 78-92).

Better to provide more literature to support those hypotheses. Usually, hypotheses are proposed based on literature reviewed.

Any justifications for choosing the five countries?

2. Materials and methods

Authors need to provide info on the list of respondents, how many emails gathered from each country, and how many responded from each country in addition to the overall response rate. How many rounds of followups were used? More info on the survey procedures is needed.

Usually, explanatory factor analysis (EFA) needs to be conducted to identify factors and then calculate reliability if those measure items are not well established in the literature. Even for established  measures, factor numbers and composition may vary with sample (indeed, most measurement scales are sample specific), and thus,  EFA is usually conducted and CFA follows.

From p. 189 on, another survey on behavior change was introduced, creating confusion with the survey done on list of respondents. The link between the two survey needs to be clarified and both need to be introduced in the introduction section. Again, CFA was used without explanation of why EFA was not used first.

p. 193 on outdoor recreation survey before and after… I guess this survey was not done by authors. If so, then need to be clarified.

p. 200 ,the collected data… which may need to be more specific. I guess this data was the data collected from the survey of respondents, April to June 2020, not from PHSM index data.  

from p. 187 on, there are three things, PHSM index, outdoor recreation survey, and the collected data, which need to be clarified a little bit more.

Discussion and conclusion

Authors need to provide theoretical and methodological implications, if any.

Author Response

Thank you for your efforts. Please see the attachment.

Reviewer 2 Report

This is an interesting study regarding international comparisons of

five African countries on the relationships of perception of health recovery, preference for urban forest, stress increase and indoor activity during the COVID-19 pandemic. I agree the significance of this study. The authors used SEM via AMOS 25 software  testing their proposed model. It is appropriate and I encourage the authors to do so. I would also encourage more cross-cultural/national research in the context of park recreation and tourism as well as African countries. The paper has the potential to specifically contribute to our understanding of the aforementioned variables from Malawi, Rwanda, South Africa, Tanzania, and Zambia perspective. In general, the manuscript is reasonably organized. However, a few points need to be further explained. My recommendations are for the authors to better frame their study and improve the presentations of the findings. Below list my suggestions in my review.

1.     AT LINE 253-254, page 7, the authors stated, "The ANOVA analysis demonstrates the differences in responses be-252 tween countries were statistically significant in all items except for preference for the ur-253 ban forest." in Table 4. However, the table 4 showed Stress increase is not statistically significant. Please also add the results on post-hoc tests in Table 4.

2.     In figure 2 at page 2, there was a double-arrow path between stress increase and preference for urban forest as well as a double-arrow path between indoor activity and preference for urban forest. Please explain why they are not single-arrow path?

3.     Please drop the CR formula at page 9 as it looks redundant.

4.     The R-square values should be demonstrated in all your SEM models tested.

5.     The study limitation needs to address the concern of comparing apples to apples in comparative studies. In other words, please add the sample homogeneity issue in a cross-national study

In sum, this study is timely and this paper addresses an important need in cross-national research. Overall, this article needs to be further enhanced with greater clarity. I suggest minor revisions to improve the quality before a further decision can be considered.

Author Response

Thank you for your efforts. Please see the attachment

Reviewer 3 Report

This study investigates the impact of the lock-downs for Covid-19 containment on visits to greenspaces and its impact on health in five countries of South and East Africa. Though impacts of greenspaces on health are common in the Global North, there are too few of them in Africa and to my knowledge no study on the impact of Covid on changing nature visitations. Consequently, this study is novel and of high interest.

However, there are unfortunately some methodological and presentation aspects that need to be addressed. Before going into their details, I would however like to precise that, though having worked with perception and use of urban greenspaces extensively, I am not totally familiar with the SEM approach used. Some of my comments might thus result from a miss-understanding of the approach, however most of the readership might find themselves in a similar situation so addressing those issues will be important nonetheless.

Methodologically, I would recommend having a small sentence, before describing each method, explaining its aims. That would greatly help the reader understanding what you're trying to achieve. At the one point where that was done, however, you mentioned that you used an Anova (and SEM) to examine the relationship between variables (L. 201-202), yet Anovas are used to analyse differences between means, not relationships. 

I also have some concerns about how some of the statements have been analysed. You correctly precised that some of the items (STA4, PR3) were reversed. However, the PR statements seem to not all go in the same direction after taking into consideration the reversal. PR1 and PR3 are positive towards use of nature and PR2 is negative. I thus wonder why PR3 and not PR2 is inversed.

Generally speaking, you also often speak about looking at differences between variables, or the research hypothesis. This makes it very difficult for the reader to understand. Please keep in mind that, even though you are deeply invested in the study, the reader might read it quickly and not necessarily remember what H3a cited 2 pages earlier is. Consequently, I would recommend being more direct about why variables, research questions and/or hypothesis you are speaking about when you mention them, throughout the manuscript.

I also think this manuscript could greatly benefit from being a bit more generalised. For instance, you always mention (including in the title) the names of the five countries. Yet speaking about Southern and Eastern Africa in the title, introduction and discussion would make it more accessible and broad. Additionally, you bring a lot of very interesting background in the discussion, but it would need to be much more linked to your study.

Additionally, find some more specific comments below:

Title: short titles have more impact, scientists scan many titles and need to be able to extract the theme in a very short time. I would suggest re-focusing it. E.g. Impact of COVID-10 lock-downs on nature connection in Africa

L. 21.               Low infections (not low infection rates)

L. 22-23.         This is a pretty strong statement. According to my knowledge, there is consensus that the amount of natural experience influences the quality of life, but not determines it.

L. 25-27.         The survey was not to investigated the structural model, it was to investigate the relationships between stress, indoor activity, perception of greenspaces, perception of health recovery and use of greenspaces. The structural model was used as analysis. Please reformulate.

L. 27-28.         Please do not mention that “The research hypothesis was supported by the results of the study”, as this does not mean anything out of context. Describe your results.

L. 37.               Covid-19 instead of coronavirus, as there are also other coronaviruses.

L. 57                diseases

L. 60-61          You describe the weaknesses of the African healthcare system and mention the low per capita healthcare cost as a weakness. Did you speak about public investment in health care? Because if that is the actual healthcare cost, I struggle to understand how having affordable healthcare is a weakness.

L. 64                Is that four beds for 100,000 people, or four beds in total?

L. 82                If you mention an acknowledged hypothesis, please describe it as it might be unfamiliar to the reader. It is more important to know what the reward hypothesis is than to know that it is the oldest.

L. 82-92          You describe here your aims, that should go at the end of the introduction.

L. 86                You write here about the “perceptions about and preferences for urban forests” yet in the rest of the manuscript, you only speak about the preferences for urban forests. Please be consistent. Also, I personally think that it would be more correct to use perceptions instead of preference, see comment about L. 138.

L. 88                What do you mean by “central factor”?

L. 106-112      This is a repetition of L. 82-91.

L. 113              Please define abbreviations (here SEM) on their first use

L. 113-121      This should go in the methods section.

Figure 1           Please define all abbreviations. An option to might be clearer would be to input e.g. “stress increase (4 statements)” without making a box for every single statement. Additionally, this figure describes your methods, not the background for the study, and should thus be in the methods section.

L. 136-149      Please reformulate as sentences. Do not hesitate to own your work and write at the first person, e.g. “we hypothesise that …”

L. 138              Throughout the article, you use “preferences for urban forests” when speaking about the appreciation. If speaking about preferences, one would think about if respondents would like e.g. more landscaped or wilder greenspaces, the inclusion of blue features etc. But not only if people appreciate/love/have positive feelings about urban nature. I would thus recommend renaming this category “appreciation of greenspace forests” (or an alternative)

L. 138-149      When citing your categories for the first time, it might be worth defining them.

L. 152              This might be a personal opinion (so I would totally understand if you don’t address it), but I always find it helpful to start the methods section with a brief overview of what is being done.

Table 1            Could there be a line for all respondents? And could you please separate the “Female/Total of respondents” into two columns: one with the number of respondents and one with the proportion of female respondents

L. 182              In which languages was the survey translated? In which language was it initially developed? Was it back-translated?

L. 183-184      What do you mean by “the reliability of each country”?

Table 2.           In additions to the points already mentioned in the general comments and on L. 138, I would rename “stress increase” to make it more neutral, e.g. “stress level”.

I would however not present this (and L. 183-184) on the methods sections, but within supplementary materials. Additionally, how were the questions quantified? You mention further down that outdoor recreation was quantified on a 5-points scale, what about those questions? Was it a Likert scale? If so with which possible answers?

L. 193-199      Please reformulate. E.g. Outdoor recreation before and after the COVID-19 lockdowns was quantified two five-points scales, for the frequency and duration of greenspaces visits.

L. 202              Anovas investigate the difference between groups, not whether there is a relation between variables.

L. 204-205      Unclear

L. 206-207      How did you “examine to avoid errors” with the collinearity? Did you exclude variables? If so, which ones? And which variables were co-linear?

L. 208-211      As previously mentioned, I am not familiar with this method. However, this information seem to me insufficient to be able to reproduce the study.

L. 215-220      This should go in the methods section

L. 221-222      “a rise in preferences” is one of the sentences that do not mean anything and would require a change from preference to e.g. appreciation. (a higher appreciation)

Table 3            First, this table is separated in two parts, the first one being a figure which should be presented as such, the second (the results of the t-tests?) as a table that could be moved to supplementary information. In the first part, the colours should be defined both within the figure and in the caption. In the second part, what do the numbers and arrows refer to? Having them properly in a table would allow to have that clearer.

L. 241              Comparison between what and what?

L. 251-254      This paragraph is not needed, referring to Table 4 and to statistics within the next paragraph should suffice

L. 253              !!! You mention all items except preferences are significant, yet the stress levels have a p-value of 0.088, which is not significant…

L. 258              Do not describe non-significant differences. You can mention that they are similar (which is a totally valid result), but not that they seem different but are not.

Table 4            It would be much nicer/easier to read as a 4 barplots (in a panel figure). This would also allow you to show where the significant differences are. Which also brings the questions of: did you carry out any post-hoc tests? You know there is a significant difference within the countries, but where does this significant difference lies? Which is general makes me wonder why you used an anova instead of GLMs, which would also allow you to include additional variables?

                        Additionally, please be consistent with the name of your categories (Health recovery vs perception of health recovery)

L. 270-271      This is a key result of the study, why not present it visually. And what about the results about the duration of greenspace visits?

L. 270-271      Reformulate: The frequency and duration of visits to the urban forests significantly decrease due to lockdown measures (T=     ; p=   ).

L 271-272       “these data did not match the research hypothesis” belongs to the discussion

L. 271-272      You results show changes in outdoor recreation, but did not investigate causality between appreciation of urban forests or perception of health recovery and outdoor recreation. Consequently, I do not understand why you state that this decrease invalidates H3. Additionally (but that might be due to my unfamiliarity with SEM), it seems to me that removing part of the diagram is misleading: if one part is deems insignificant, it should be illustrated as not being linked, shouldn’t it?

Fig. 2               In additions to my comments regarding the readability from Fig. 1, I would not recommend repeating the figure. One can show e.g. with colours or shapes what has further been disproved.

L. 286-330      I do not understand what you are trying to do with this section, nor do I think it was explained in the methods. If this is commonly part of the SEM, I would recommend explaining it clearer for anyone not familiar with the method, especially that SEM are usually used in psychology and Land is an environmental journal, focusing on landscapes, biodiversity, etc. so it is likely that most of the audience is unfamiliar with the methods. However, this section (including Table 5) did not seem to me to add much to the results and could be transferred to additional information.

L. 336              You did not mention using chi-squares in your methods. Also, from what I understand, you did quite a lot of comparisons (see table 6): did you compensate for multiple tests e.g. through the Bonferroni correction?

                        Additonnally, while presenting the results of a chi-square, the results is not “acceptable”: please reformulate e.g. there were no significant relationship between the stress level and the perception of urban forests (values).

L. 337-338      What does the first number refer to? The chi-value? If so please state it.

Table 6            Please be clearer in column 1, e.g. “Relationship between” as a column title and “stress level and perception of urban forest”, etc. as a line names

                        I would recommend just showing accept/reject, and putting the values in supplementary information

                        Please add the comparisons for all countries together, so that the results are more generalizable.

L. 364-365      This goes into the methods

L. 379              This table does not show the effect of the Sobel tests, it shows the indirect effects of explanatory variables on heath outcomes, evaluated through the Sobel test

Table 7            With a clearer table legend, you could just add the primary variable (stress/indoor) in the 1st column.

L. 382              Maybe start the discussion with a brief summary of the results?

L. 384-385      Remove the parenthesis

L. 392              after the pandemic declaration

L. 392              Please cite the countries

L. 395-399      this is introduction material. Lines 383-394 probably as well.

L 400               Own your results, be proud of your work!! “We found increased stress levels due to…”. This also makes it more readable.

L. 403              do not refer to tables within the discussion (though they should all be referred to within the result section). Valid throughout.

L. 410-411      remove “which showed…”

L. 426              replace “verified with a high significance levels” by “observed”

L. 431              Reformulate: We predicted that…

L. 434              A hypothesis does not establish anything, it is a way of looking at the data. Reformulate

L. 439              remove “of the mediating effect verification”

L. 441              heath recovery, not promotion

L. 446              Define your abbreviations at the first use.

L. 447              Alpha wave might need to be defined regarding the readership of the journal.

L. 469-494      Not needed as such in the discussion, though some elements might be included in the methods or added to some parts of the discussion.

L. 500              precise which pandemic

L. 501              This is the time to generalise: “in most but not all of the investigated countries”

L. 515-520                  What do you mean by “software” and “resources” as an author contribution? And should not have all the co-authors have given some review on the manuscript?

References      A lot of references! A very good background about covid but all this research is new and has been published very fast. What about adding a bit more on the heath and well-being impacts of greenspaces in the Global South? Suggestions:

            Nawrath et al. 2021 Environmental Research

            Tomita et al. 2017 The Lancet Planetary Health      

Author Response

Thank you for your efforts. Please see the attachment

This manuscript is a resubmission of an earlier submission. The following is a list of the peer review reports and author responses from that submission.